# The Legacy of Prince Khaemwaset at Saqqara

Campbell Price

Manchester Museum, University of Manchester, Manchester M13 9PL, UK; campbell.price@manchester.ac.uk

**Abstract:** Saqqara in particular, and the Memphite necropolis in general, constituted the arena for the prolific and significant monumental self-presentations of Prince Khaemwaset, fourth son of King Ramesses II (c. 1279–1213 BCE). The present paper explores the role of the prince in fashioning a persona that addressed past, present and future audiences. This discussion is used to contextualise results of the 2009 Saqqara Geophysical Survey Project, showing the greater-than-expected extent of the New Kingdom necropolis south of the Unas Causeway. It considers responses to the deep palimpsest of the sacred landscape of the Memphite necropolis by—and later commemorations within it of—this notable Ramesside individual.

**Keywords:** Egypt; Saqqara; Khaemwaset; Ramesside; archaism; Late Period; commemoration

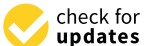

## 1. Introduction

The figure of Khaemwaset, the fourth recorded son of Ramesses II (c. 1279–1213 BCE) and High Priest of Ptah at Memphis, is well known from his many inscribed monuments [1,2]. Based especially on the propensity of the prince and his agents to add secondary inscriptions to standing monuments, he has frequently been praised and mythologised as a proto-Egyptologist [3] (pp. 78–96). Khaemwaset's apparently exceptional status as an 'antiquarian' ought, however, to be seen in the context of his own time, while his legacy at Saqqara should be considered in light of some later archaeological sources that have previously been overlooked or misinterpreted.

Of any of his (many) siblings, Khaemwaset was certainly the most prolific in perpetuating his own name and image. The obvious exception was his brother, Merenptah, who eventually succeeded their father; both kings are well-known for adapting and adding inscriptions to earlier monuments, especially sculpture in the round [4] (pp. 30–35). As a deep palimpsest, the site of Saqqara in particular and the wider Memphite necropolis in general provided an arena for Khaemwaset's monumental self-presentation. Here, I discuss aspects of that presentational strategy, his relationship with the built landscape, his forebears and the echoes of his presence at Saqqara centuries after his death.

## 2. Princely Precedents

A major reason for the heroic status of Khaemwaset amongst Egyptologists today is the literary fame he achieved by Ptolemaic times as a quasi-historical character in the Demotic 'Setne' tales. The stories play out largely within the sacred landscape of the Memphite necropolis itself, while the central narrative of the first tale concerns Setne's search to find the tomb of a much earlier royal son, Naneferkaptah, of whom it is said:

> '[(And so) it happened that Nanefe]rka[ptah] my brother [had no] occupation in the land aside from hiking over the necropolis of Memphis, as he recited the writings that were in the tombs of the pharaohs, along with the stelas of the scribes of the House of Life, as well as the writings that were on [their tombs. How] very much [he rejoice]ˉdˉ(??) because of writing!' [5]. (p. 114)

That this 'occupation' was thought to be something eccentric by the Ptolemaic Period is implied by the laughter it provokes from an old priest, who believes it to be useless. An

intriguing historical parallel for seeking out the material remains of a forebear comes in the form of a seated statue of Kawab, an elder son of King Khufu of the Fourth Dynasty, apparently found at Mit Rahina [1] (pp. 67–69, 84) [6] (pp. 470–473). An extensive secondary inscription on the seat of the statue records the action of Khaemwaset in removing it from what was 'cast (away) (Sdyt) in [ . . . ] of his father, King of Upper and Lower Egypt, Khufu [ . . . ] decreed [ . . . ] place in Memphis, (he) libates the gods in the company of the excellent akh-spirits before the chapel (Hwt-kA) of Ro-Setjau' [6] (pp. 471–472).

It seems clear that the statue's original context was Kawab's sizeable tomb chapel in the Eastern Cemetery at Giza, which was apparently deliberately denuded not long after construction [7] (pp. 193–194). The reported Mit Rahina provenance suggests that the already fragmentary sculpture had been re-dedicated in the temple of Ptah at Memphis—Khaemwaset's main focus of operations as High Priest of Ptah—in what Snape [7] (p. 193) terms a 'statue park'. Certainly, the existence of a collection of statues of kings of the Old Kingdom at Mit Rahina may suggest a comparably plausible context [8] (pp. 842–843). But the specific mention of a 'ka-chapel of Ro-setjau'—a toponym that designates the desert area stretching from Giza to Saqqara [9] (p. 271)—rather calls to mind the prominent structure at North Saqqara excavated since the 1990s by a Japanese–Egyptian mission, which itself is described as a 'ka-house to the west of Memphis' in a title held by a scribe named Pentawer in an ostracon inscription found at the site [10] (p. 166).

Khaemwaset's chapel—for there is no firm evidence that it served as his tomb—is located on a hill and enjoys an impressive set of views of Old Kingdom monuments throughout the Memphite necropolis. A cache of statuettes—of various forms, including lioness-headed goddesses and sphinxes—naming Khufu and Pepi I found within the structure [11] (pp. 26–29) illustrate a crucial and repeatedly asserted connection between Khaemwaset and his royal forebears—the sort of 'excellent akh-spirits' with whom the prince would wish to keep eternal company. Given the extensive reuse of Old Kingdom masonry in this Saqqara 'ka-house' [12] (pp. 576–577), perhaps this was the initial or intended setting for Kawab's re-inscribed statue.

The singularity of the historical Khaemwaset's actions should be seen against a general background of engagement with the monumental past displayed by other Ramesside elites. Visitors' graffiti are commonly attested from throughout the New Kingdom, especially in the Memphite necropolis [13], and the few surviving examples of what we term 'king lists' all date to the Ramesside Period [7] (pp. 188–190). More specifically, in terms of Khaemwaset's own social milieu, a man named Pahemnetjer was an almost immediate predecessor as High Priest of Ptah at Memphis and had 'antiquarian' interests of his own, judging by 'restoration' inscriptions for some of his predecessors found on their monuments [14] (pp. 144–145, 277, 288).

The association of earlier princes with their ancestors' monuments provides further precedents. An 18th Dynasty royal son named Nakht is known from a statue found in the Karnak Cachette to have discharged the function of ka-priest for statues of Ahhotep, Tuthmose I and Hatshepsut [15] (pp. 47–48) [16]. A series of stelae dedicated by a royal son or sons of Amenhotep II at Giza indicate that the Sphinx and its enclosure were a site of royal filial piety, presaging claims to such devotion made on Tuthmose IV's famous 'dream stela' [17] (pp. 187–192).

In this connection, Khaemwaset's 'ka-house' at north Saqqara is built right next to a significant structure from the reigns of Amenhotep II and Tuthmose IV. The existence of a common axis and possible reuse of masonry make it highly likely that Khaemwaset's builders depended on the earlier structure for inspiration and for building material, while the arrangement of stelae of Tuthmose IV within its associated mudbrick walls [10] (p. 164) recalls the suggested arrangement of stelae of the same period within the Sphinx enclosure at Giza [17] (p. 196). Intriguingly, a monumental offering basin dedicated by Khaemwaset to Imhotep 'son of Ptah' and very likely from Saqqara carries an inscription referencing a 'divine enclosure' (stpt-nTr)—employing the same term (stpt) used of the Giza Sphinx enclosure in ancient sources [18] (pp. 8–9) [19] (pp. 137–138). The strong implication is of a

common theme of royal ancestral memorialisation at significant cultic sites in the Memphite necropolis [12] (p. 578).

Khaemwaset is well known for building work at the Serapeum complex, located just less than a kilometre to the south-east of his north Saqqara chapel. Discussion of his connection with the Serapeum has tended to focus on the identity of a putative 'mummy' associated with the prince; it now seems clear that this masked, wrapped form was associated with the Apis bull [20] (p. 115). Khaemwaset was likely responsible for the initiation of the so-called 'Lesser Vaults' in his father's year 30, connected with the king's first heb-sed [21] (pp. 79–84) [20] (pp. 113–123). It is interesting to note the role apparently played by another eldest prince, Tuthmose son of Amenhotep III, as founder of the so-called 'isolated tombs' for the Apis bulls [20] (p. 117), setting yet another precedent for Khaemwaset [3] (p. 89).

When the Serapeum was first extensively excavated in 1850s, workers for Auguste Mariette (1821–1881) found some 50 'shabti' figurines—most inscribed for Khaemwaset—in the sarcophagus of the Apis bull that died in year 30 [22] (pp. 230–299). Any simple interpretation of these as connected to a putative burial of Khaemwaset is misguided; such extra-sepulchral figurines ensured the presence and participation of the individual represented at a given sacred site [23].

During excavations directed by W.M. Flinders Petrie (1853–1942) south of the pyramids of Giza at Kafr el-Batran, many such votive 'shabtis' of Khaemwaset were found [24] (p. 24). Petrie noted at the time that there was no suggestion that the prince had been buried there, later even terming this site 'a cenotaph of Khoemwas' [25] (p. 206). Petrie corrected an erroneous claim by Gaston Maspero to have identified the owner of a nearby tomb as the son of Ramesses II Khaemwaset—rather than the real owner, Master of the Royal Wig of the same name, of apparently New Kingdom date [8] (p. 304). The proximity of so many figurines to the tomb of a like-named individual may, perhaps, be no coincidence. Might Khaemwaset have been attracted by the name—or might that individual have known of Khaemwaset's deposit?

Many more such votive figurines exist naming Khaemwaset from the Memphite area generally [26] (pp. 16–22), with more elaborate examples provided with the so-called 'Khaemwaset formula' used on his and other 'shabti' figures of early Ramesside individuals. This formula thematises the Memphite ritual landscape, explicitly situating the transfigured deceased within the 'sacred land' of Ro-stejau and emphasising the gift of sight [9] (pp. 269–271). More than any other known ancient individual, Khaemwaset appears to have had knowledge of the significance of—and keenness to integrate his own name within—the Memphite necropolis and its visible features.

### 3. Mobilising the Monumental Past

Based on the surviving evidence, it is difficult to avoid the impression that Khaemwaset had a personal agenda that related closely to the monumental vestiges of previous rulers and elites. This was not an abstract interest in the past; his recorded activities had a material focus, evidencing interaction with accessible monuments he connected with his own ancestors, a genealogy in stone that chimed in with broader Ramesside high cultural desire to account for and associate with esteemed forebears. Rather than simply a 'scholarly prince'—reconjured in the image of so many (Western) Egyptologists—Khaemwaset framed his actions as an agent for his father while also able to present himself as an individual [6] (pp. 470–473).

Although Khaemwaset's asserted pride in intellectual enquiry is shared by contemporary royal inscriptions both in Egypt—as, for example, in his father's 'researching in archives' recorded at Luxor Temple [27] (p. 144), but also elsewhere in the ancient Levant [28] (p. 74)—he was motivated by expectations of active reciprocity that are particularly pronounced in non-royal, as opposed to kingly, monumental discourse [29] (p. 29). Khaemwaset's actions towards his forebears—including his own father—were undertaken in the expectation that future generations would act in turn for him. The previously men-

tioned statue of the Fourth Dynasty Prince Kawab specifically vaunts a motivation for Khaemwaset's actions:

'so greatly did he love antiquity (pAwt) and the noble ones who came before (Spsw imyw-HAt)' [6]. (pp. 471–472)

The rhetoric of reciprocity is clear; he describes his love of those 'who came before' for those who will come in future—as his dedicatory inscription in the Serapeum makes explicit [2] (pp. 91–92). Without the assured divinisation that came of being king, which was perhaps never a role he expected to take on, Khaemwaset had to try particularly hard to make a mark on the monumental record to inspire future generations to act accordingly for him.

Thus, the offering basin he had dedicated to the divine Imhotep [18] implies that there were sources available to Khaemwaset that identified Imhotep as an exceptional individual connected to building activity at Saqqara. And it was in Imhotep's footsteps as an agent of royal action that Khaemwaset perhaps most wanted to follow, aligning himself specifically with a fellow non-king who received veneration as a worthy monumental actor after his own death. The fact that the basin is the first known attestation of Imhotep's epithet 'son of Ptah' raises the very intriguing possibility that Khaemwaset himself had a key role in launching the sage's cult.

The magnified status of Khaemwaset as quasi-royal patron is indicated by the appearance of his name and title in place of those of the king in an inscription on a set of canopic jars of a man named Ty, very likely from Saqqara [30] (pp. 399–407) [2] (pp. 47–48). In an apparently unique variation of a standard formula asserting royal favour [31] (pp. 818–822), it is Khaemwaset who is said to dispense the material gift of the finely carved alabaster jars as Hswt ('favour', 'gift'). The use of the formula itself, which otherwise rarely appears during the Ramesside Period, is a conspicuously archaising deployment of text [32] (pp. 398–399)—but also a way of mediating a royal role, as in the monumental 'labels'—that formed a key assertion by Khaemwaset and his agents.

## 4. Geophysical Results and 'Labelling' Monuments

Employing methods developed in Saqqara since 1990, the Scottish–Egyptian Saqqara Geophysical Survey Project has been able to build a subsurface map that has greatly aided the interpretation of visible structures at the site. Using a Fluxgate gradiometer FM 256 to walk 30 by 30 metre squares, located on a GPS grid [33] (pp. 79–82), work has followed standard methodologies adopted in archaeology [34] (pp. 66–69). Due to the contrast between mudbrick architecture, which holds a magnetic charge, and the sands of Saqqara, structures show up very clearly on the resulting geophysical plot [35,36]. Disturbances or anomalies on the plot are chiefly caused by metal cables used to provide lighting to other parts of the site.

In 2009, the project, directed for its last season by the late Ian Mathieson (1927–2010) and accompanied by the writer, extended the area of its survey to the south of the Unas causeway (Figure 1). The results shown below were gathered in the area of a significant New Kingdom cemetery, best known as the site of the impressive 'temple-tomb' of the army general Horemheb, who eventually succeeded Tutankhamun as king [37].

Horemheb was ultimately to be buried in the royal cemetery at the Valley of the Kings, but it appears likely that his earlier tomb at Saqqara became the nucleus of a much larger New Kingdom necropolis there. Excavations have been undertaken by an Anglo-Dutch mission from the 1970s onwards [38], along with Cairo University in the 1980s and more recently [39]. These have revealed a number of Ramesside structures nearby and the existence of more was to be expected in this area [40] (pp. 508–513). Museum collections around the world contain many 'Memphite'-style limestone reliefs, characteristic of New Kingdom Saqqara. These reliefs probably derive from the site, although most of their original structures have long been covered with sand. Crucially, the internal walls of such 'temple-tombs' are made of mud-brick and so were ideally suited to detection by gradiometery. Indeed, in 2009, surveys of the area around Horemheb's tomb revealed many

more neighbouring structures, all roughly aligned east–west and often showing one or more 'courtyards'—both features that would be expected during the Ramesside period.

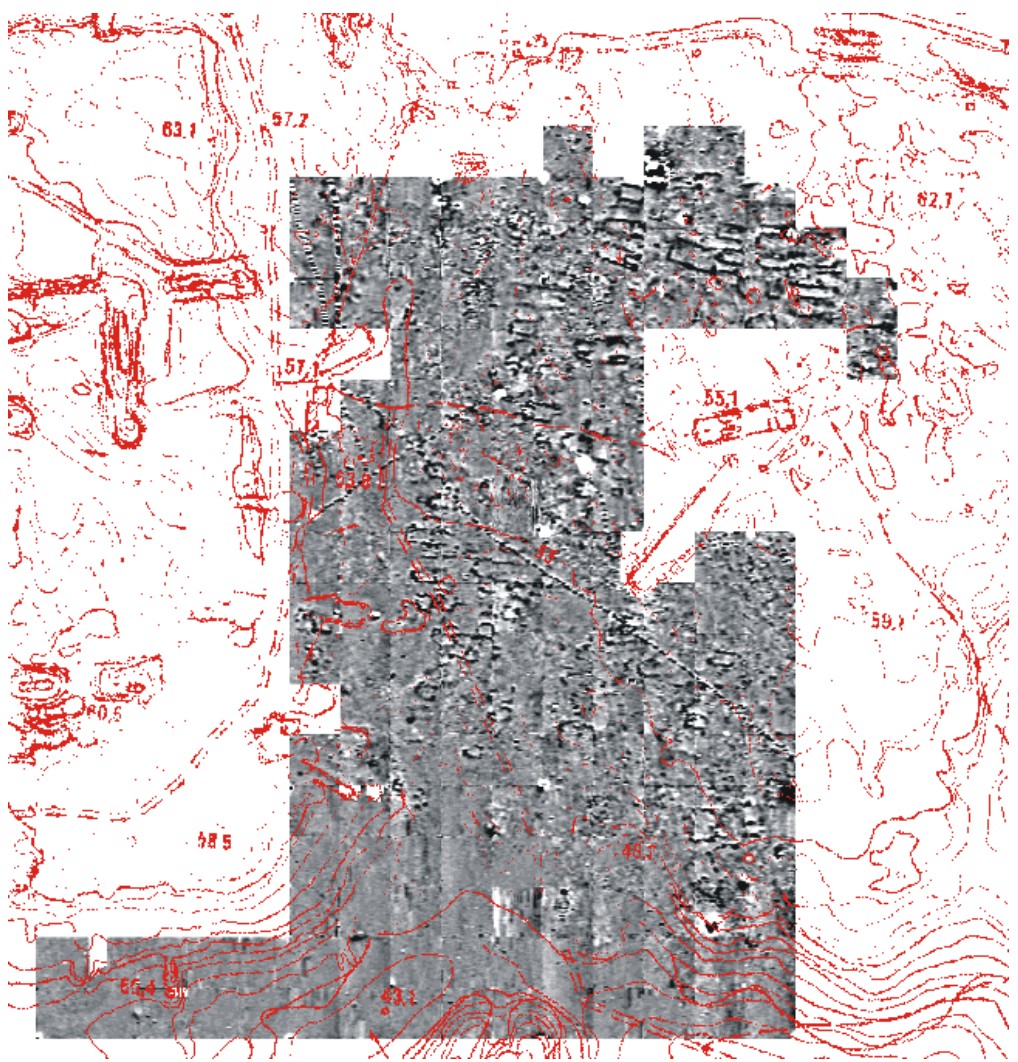

**Figure 1.** Results of the Saqqara Geophysical Survey Project 2009 season, south of the pyramid and causeway of Unas, showing clusters of east–west oriented New Kingdom tombs stretching west of the tomb of Horemheb. The red contour map overlaying the geophysical plot comes from the 1978 Egyptian Ministry of Housing and Reconstruction.

The geophysical survey made clear that these structures extended further west than expected. In this context, the writer wondered if the extent of this cemetery might elucidate the choice of location of a secondary inscription in the name of Khaemwaset on the southern side of the pyramid of Unas, facing the contemporary tombs.

Distinctive markers of Khaemwaset's activities are the inscriptions in the names of he and his father added to several standing monuments in the Memphite necropolis and at other sites. These texts are known both from surviving examples but may in some cases be attested indirectly from later accounts of the appearance of monuments. The Unas pyramid inscription was found on its south side [41], and is now restored in that position. It is the best preserved of such texts and is typical in formulation, being framed by a royal command:

'His Majesty decreed an announcement; It is the High Priest (of Ptah), the Sem-priest, King's Son Khaemwaset, who has perpetuated the name of King [Unas]. Now his name was not found on the face of his pyramid. Very greatly did the

Sem-priest, King's Son Khaemwaset, desire to restore the monuments of the
Kings of Upper and Lower Egypt, because of what they had done, the strength
of which was falling into decay. He set forth a decree for its sacred offerings, . . .
its water . . . [endowed] with a grant of land, together with its personnel . . . '.
(writer's translation)

Other such inscriptions appear in different positions on individual monuments: on
the north side of the mastaba of Shepseskaf [1] (p. 77, no. 12); the south side of the Step
Pyramid [1] (p. 77, no. 8); the east side of the pyramid of Userkaf [1] (p. 77, no. 9); the
south side of the sun temple of Niuserre at Abu Ghurab [1] (p. 76, no. 4); an unknown
location on the pyramid of Sahure at Abusir [41] (pp. 205–207); and on the south side of the
pyramid of Pepi I at Saqqara [42].

Blocks with the tell-tale phraseology of Khaemwaset's monumental calling-card were
found to the south and south-east of the pyramid of Senwosret III at Dahshur, with the
interesting indication that there were several different iterations of the same inscription [43]
(pp. 29–30).

In addition to these archaeologically attested examples, the 1st Century BCE Greek
historian Diodorus Siculus mentions [44] (64.3) that the name of Menkaure is to be found
on the northern face his pyramid. Herodotus also refers to an inscription on the Great
Pyramid but does not specify on which side it appears [45] (II, 125). The Arabic scholar Ibn
Khurradadhibih (d. 911 CE) mentions the pyramids of Giza generally being inscribed [46]
(p. 72), while Edward Lane (1801–1876) also refers to the common assertion in Arabic
sources of the 9th to 13th Centuries CE, with which he was familiar, that the two principal
pyramids bore 'numerous inscriptions, in unknown characters', but he assumes these to
have been in Greek [47] (pp. 196–197).

Several inscribed monuments and votive deposits attest to Khaemwaset's activity
at Giza [19] (pp. 201–206). Undoubtedly, the removal of the outer casings of the Giza
pyramids during the Middle Ages would deprive us of any evidence of his secondary
inscriptions on them [48] (p. 42). Thus, by the earlier 18th Century CE, the Danish traveller
Frederik Norden (1708–1742) noted with surprise the complete lack of inscriptions inside
or outside the Giza pyramids [49] (pp. 44–45).

Finally, it is also worth acknowledging the assertion by Herodotus [45] (II, 148) that
the pyramid of Amenemhat III (at Hawara) has 'large figures carved upon it'. While this
may be a reference to the inscribed capstone [50] (p. 263), it seems unlikely that this text
could have been seen when in situ, and perhaps this represents an allusion to yet another
inscription by Khaemwaset.

The suggestion by Snape [6] (p. 470) that sight-lines with the temple of Ptah at
Memphis may have influenced the placement of Khaemwaset's additional inscriptions
is intriguing and raises the question of their deliberate orientation. Drioton [41] (p. 206)
speculated on the significance of placement, arguing for the primacy of a monument's
southern face. He pointed out that the eastern side was usually the location of the pyramid
temple: 'and the inscription would have had to be carved too high for it to be read. As for
the west face, it was practically out of sight' (writer's translation from the French). However,
examination of the surviving examples does not fully corroborate Drioton's theory, nor
does it imply a firm pattern of occurrences (Table 1), although it is worth considering again
the purpose of Khaemwaset's 'labels' and their placement.

That these are 'restorations' in the sense of modern heritage management is called
into question by, for example, the 'restoration inscriptions' of Ramesses II at Deir el-Bahri,
which are associated with vandalised and apparently un-restored texts and scenes [51]
(pl. 52). Claims that Khaemwaset's assertions were a front for quarrying material [52]
(pp. 65–66) may also overly simplify the picture; interpretations of piety and expediency are
not mutually exclusive, and intentions may be nuanced [7] (p. 194). That these secondary
inscriptions may in some way have compensated for the removal of inscribed stone from
adjoining pyramid complexes [50] (pp. 264–265) also seems plausible and—importantly—
takes into consideration both a living and an eternal audience. Such texts certainly effected

the presence of both Khaemwaset and his father on the monument of a previous king. Crucially, the texts allowed Khaemwaset to participate in these kingly monuments for eternity—following the concept of 'spiritual participation', exemplified by Middle Kingdom reuse of masonry blocks from Old Kingdom royal mortuary complexes, suggested by Hans Goedicke [53] (p. 6).

**Table 1.** Secondary monumental inscriptions of Khaemwaset.

| Location | Position | Attestation |
|---|---|---|
| Pyramid of Unas, Saqqara | South side | Survival |
| Mastaba of Shepseskaf, Saqqara | North side | Survival |
| Pyramid of Djoser, Saqqara | South side | Survival |
| Pyramid of Userkaf, Saqqara | East side | Survival |
| Temple of Niuserre Abu Ghurab | South side | Survival |
| Pyramid of Sahure, Abusir | ? | Survival |
| Pyramid of Pepi I, Saqqara | South side | Survival |
| Pyr. of Senwosret III, Dahshur | South side (?) | Survival |
| Pyramid of Menkaure (?), Giza | North side (?) | Reference |
| Pyramid of Khufu(?), Giza | ? | References |
| Pyr. of Amenemhat III, Hawara | ? | Reference |

In the case of the pyramid of Unas, I suggest that in addition to this general sense of eternal monumental participation, it was the proximity of the extensive New Kingdom cemetery, now demonstrated by geophysical evidence, that influenced locating the inscription on the pyramid's southern side—targeting an audience among Khaemwaset's recently deceased and divinised ancestors, and those who came to visit them. Khaemwaset certainly made play with the age-old 'appeal to the living' formula elsewhere at Saqqara, with a special dedicatory inscription entreating temple staff permitted entry into the sacred precincts of the Serapeum to remember and act for him [2] (pp. 91–92).

Khaemwaset's ritualised concern for particular cardinal directions is strongly implied by the text of the libation basin he dedicated to Imhotep, which very likely originates from Saqqara [18] (pp. 1–10). In it, Khaemwaset also addresses very specifically 'the living god(desse)s of the south' and 'gods of the south and gods of the west' [18] (pp. 2–6). Although the editor of the text suggested that these 'gods (nTrw)' refer to the royal dead of south Saqqara known from their monumental tombs [18] (p. 9), the so-called 'Harper's Song of King Intef', attested from to the late 18th and early 19th Dynasties, may blur the distinction between kings and non-royal elite dead:

... nTrw xprw Xr-HAt Htp m mr=sn saHw Axw m-mitt qrs m mr=sn qd Hwt nn-wn st...

... the gods who existed before rest in their tombs, the effective ancestors likewise buried in their tombs; the chapel-builders, (their?) places do not exist ...

This canonical example of the 'Harper's Song' genre even invokes the example of Imhotep in the context of his tomb having become unidentified, and the composition is in fact attested in its fullest inscribed form in the chapel of Paatenemheb from the New Kingdom necropolis at Saqqara [54] (pp. 107–114), likely located somewhere south of the Unas causeway. Fearing the same fate as Imhotep, perhaps, Khaemwaset sought prominence among his contemporaries and eternity. The Unas pyramid inscription, described by Kenneth Kitchen [55] (p. 109) as the 'biggest museum label in the world', may indeed have been as much an 'appeal to the living' as the texts that were more explicitly framed in that form; vaunting his effectiveness, billboard style to those buried in that significant Saqqara graveyard and those living people who would have come to interact with the dead in that setting.

## 5. Iconography of the Iunmutef

An important role that succinctly expressed more than any other Khaemwaset's attitude to monuments—and the divinised entities whose presence they effected—was that of Iunmutef ('pillar-of-his-mother'), a semi-divine figure who had a priestly function primarily in relation to the (divinised) king rather than to full deities [56] (p. 85). Significantly, the prince asserts the role of Iunmutef when venerating Imhotep in his earliest attestation as divinised 'son of Ptah' [18] (pp. 2–3, 8). To some extent, it might even be argued that Khaemwaset refashioned and helped popularise this role.

The iconography of the Iunmutef epitomises a filial position, with the sidelock associated with youth and the panther skin of the sem-priest, the prince's other most common title. Khaemwaset is explicitly designated as 'Iunmutef' many times in enumerations of his titles [1] (pp. 23–24), including on his Saqqara monument [57] (p. 411). In this role, he performs not simply as a biological royal son but as an officiant of the divinised Ramesses II, or perhaps particularly of the royal ka-spirit in some way enhanced by the performance heb-sed rituals from regnal year 30 onwards [58] (pp. 260–261).

The potency of the royal cult whilst the king was still alive is best illustrated by individually named 'cult colossi' [59] (pp. 403–411) which invited and expected royal and divine attention and which received active veneration throughout Ramesses II's life. Khaemwaset is very likely depicted as Iunmutef on the base of at least one colossal statue of his father from the Ptah temple at Memphis [52] (p. 66, Figure 2) and in the negative space between the legs of others. This qualifies the divine king as worthy of semi-divine veneration, while emphasising the quasi-divine role of Khaemwaset himself—perhaps the only avenue to divinisation appropriate for a living son of the king.

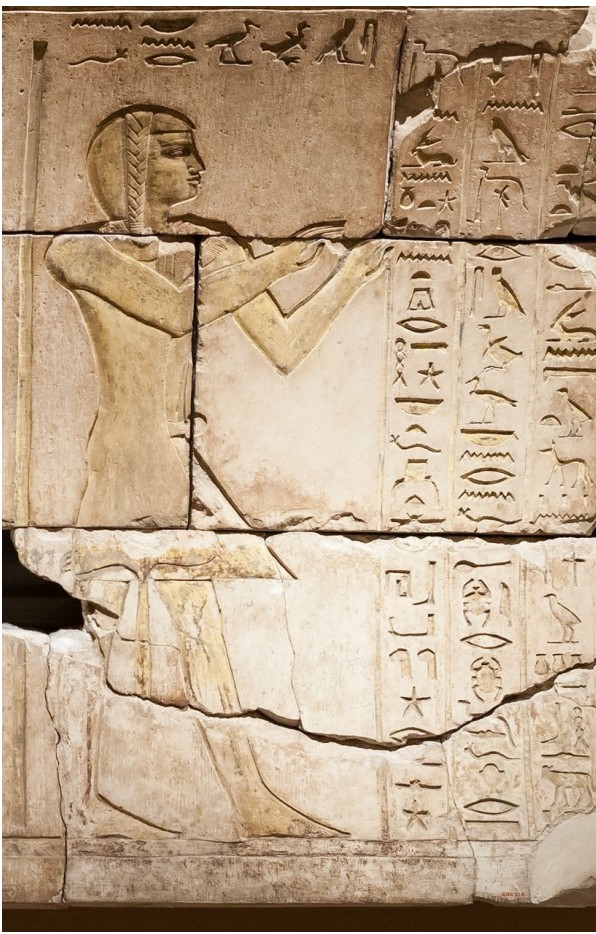

**Figure 2.** Detail of ceiling section from the 26th Dynasty tomb of Bakenrenef at Saqqara, showing the archaising iconography of the Sem-Iunmutef. Metropolitan Museum of Art 11.150.50b1.

The number and conspicuousness of Khaemwaset's depictions in relation to his father made him to some extent synonymous with the role of Iunmutef for later observers. It is notable at Saqqara, for example, to witness the Vizier Bakenrenef take on the title of 'sem' and Iunmutef, displaying the short wig and sidelock in his impressive subterranean tomb [60] (pp. 15–38) (Figure 2) and on statuary [61] (pp. 396–405). This is despite the fact that Bakenrenef did not discharge the role of High Priest of Ptah (although a different man of that name held that title [14] (p. 76, n. 1)). Even within the generally archaising context of Saite elite visual culture, at Saqqara, in particular, such a choice may suggest a deliberate evocation of the iconography of Khaemwaset. Furthermore, the rather selective and ostentatious use of short wigs in some examples of Late Period northern sculpture may have owed more to Khaemwaset's depictions than to any more generically old-fashioned forms [29] (p. 28 and n. 12).

Given the intense (re)use of spaces at Saqqara in the Late and Ptolemaic periods in connection with a series of cults, many associated with sacred animals, it is surely no surprise that Khaemwaset and his distinctive iconography were remembered and emulated there.

## 6. Posthumous Legacy

For the historical Khaemwaset to have played such a significant role in the Ptolemaic Setne tales, a connecting tradition of almost a millennium of posthumous recognition must be presumed. As is well known, Ramesses III had a son called Khaemwaset, part of a clear emulation of the naming patterns of Ramesses II's children. Somewhat later, the inclusion of the element 'Kha-em-waset' ('appearing-in-Thebes') in the birth names of both Ramesses IX and XI [62] (pp. 133–135) may not be simply a geo-political acknowledgement of the southern city but rather a reference to the son of Ramesses II [63] (p. 192) [3] (pp. 91–92).

During the 22nd Dynasty, a High Priest of Ptah at Memphis named Shedsunefertem appropriated a very fine kneeling statue of Khaemwaset, motivated less out of expediency and surely more from a desire to harness the material power of his illustrious predecessor [64] (pp. 182–183). Khaemwaset has been identified as the owner of two fragmentary statues that were dated to the Late Period by Ludwig Borchardt [65] (p. 107, (CG 1205), p. 112 (CG 1213)), but these have not been possible to corroborate.

The content of the Demotic Setne narrative finds an interesting contemporary parallel in 'find notes' to Book of the Dead spells 166 and 167, known from at least two manuscripts of the Ptolemaic Period, which attribute their discovery to Khaemwaset, 'under the head of a blessed spirit [Axw scil. a 'mummy'] to the west of Memphis' [66] (pp. 227–228 and n. 106)—a localisation within the Memphite necropolis, if not specifically at Saqqara itself. In this context, it is of significance that Khaemwaset's ka-chapel at Saqqara was the site of considerable activity during the Late Period, attested by the discovery there of quantities of votive pottery, faience vessels and bronzes; the original wall reliefs even attracted Greek graffiti that have tentatively been dated to the Ptolemaic Period [10] (p. 167).

Perhaps of greatest importance in understanding Khaemwaset's posthumous legacy is a piece that has been often overlooked and frequently misinterpreted. It is a limestone block, now in the Egyptian Museum (Special Register no. 11735) (Figure 3), apparently first published by Farouk Gomaà in his compendium of Khaemwaset's monuments as contemporary with the prince [1] (92, cat. No. 88 and Pl. Vib). The block has subsequently appeared in several publications, always being assigned a New Kingdom date [67] (pl. CIX, fig 241) [14] (p. 330, no. 142) [68] (p. 106 and pl. 137a). Only Kenneth Kitchen, in his indispensable *Ramesside Inscriptions*, seems ever to have raised the possibility in print that the piece was later—representing, as Kitchen speculated, a 'commemoration in Late Period(?)' [69] (pp. 892–899) [70] (p. 580)—but this comment has apparently provoked no further discussion of the piece.

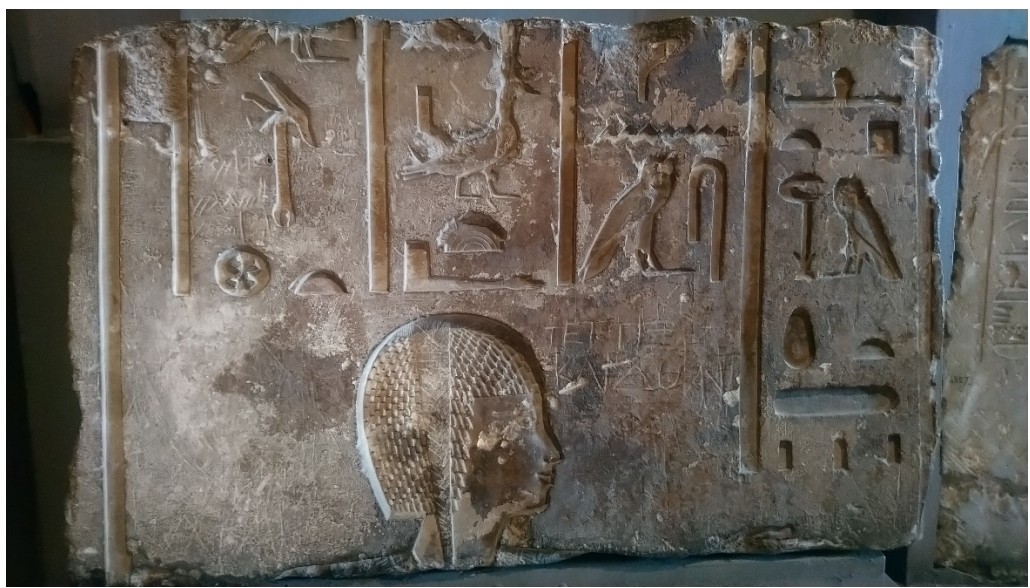

**Figure 3.** Block showing Khaemwaset. Egyptian Museum, Cairo. Special Register no. 11735.

Both the style of the relief and the palaeography of the hieroglyphs strongly suggest that it must date to the 30th Dynasty or early Ptolemaic Period. The face has the small, pert features of the Saite Period and archaistic later emulations of it; the globular shape of the short wig, with its slightly curved echeloned sections, compares, for example, with the 30th Dynasty 'Dattari' statue in Boston [71] (101, pl. 76); its lower line ends well above the shoulder unlike other Ramesside examples of the wig in relief [2] (pp. 72–73) [72] (p. 102).

Although the relief is apparently without firm provenance, Gomaà suggested Memphis or Saqqara for its origin and Fischer [68] (p. 106) attributes it to Saqqara. Its location on display in Cairo Museum with other reliefs from the Monastery of Apa Jeremias at Saqqara may provide circumstantial evidence for a common find-spot. James Quibell's excavations at the Monastery included fragments naming Khaemwaset, many associated with fragments bearing cartouches of Nectanebo I [73] (p. 4). Likewise, a fragment of a false door naming Khaemwaset was found during 1971 excavations near the Serapeum along with Late Period and Ptolemaic papyri and other objects [74] (p. 155), suggesting an awareness of who Khaemwaset was at First Millennium BCE Saqqara.

Gomaà, in particular, was resistant to suggestions that objects carrying the prince's name and titles name might post-date Khaemwaset's lifetime [1] (p. 74). While the Cairo relief may represent a much later replication of a New Kingdom scene, as is known from Theban temples in Ptolemaic times [75] (pp. 69–96) [76] (pp. 3–4), it seems rather more plausible that this was a new construction in honour of Khaemwaset.

Quite apart from the likely visibility of Khaemwaset's secondary inscriptions on standing monuments of the Old and Middle Kingdoms, extensive building work at and near the Serapeum under Nectanebo I and II is well attested [77]. That such construction or renovation at Saqqara might have included dedicated cultic space for Khaemwaset seems eminently reasonable. The identification and limited excavation by the Saqqara Geophysical Survey Project of mudbrick temple structures oriented directly towards the Serapeum [33] (pp. 86–89), identified by associated pottery as being of Late and Ptolemaic date [78], could have provided a plausible original context for such a relief (Figure 4).

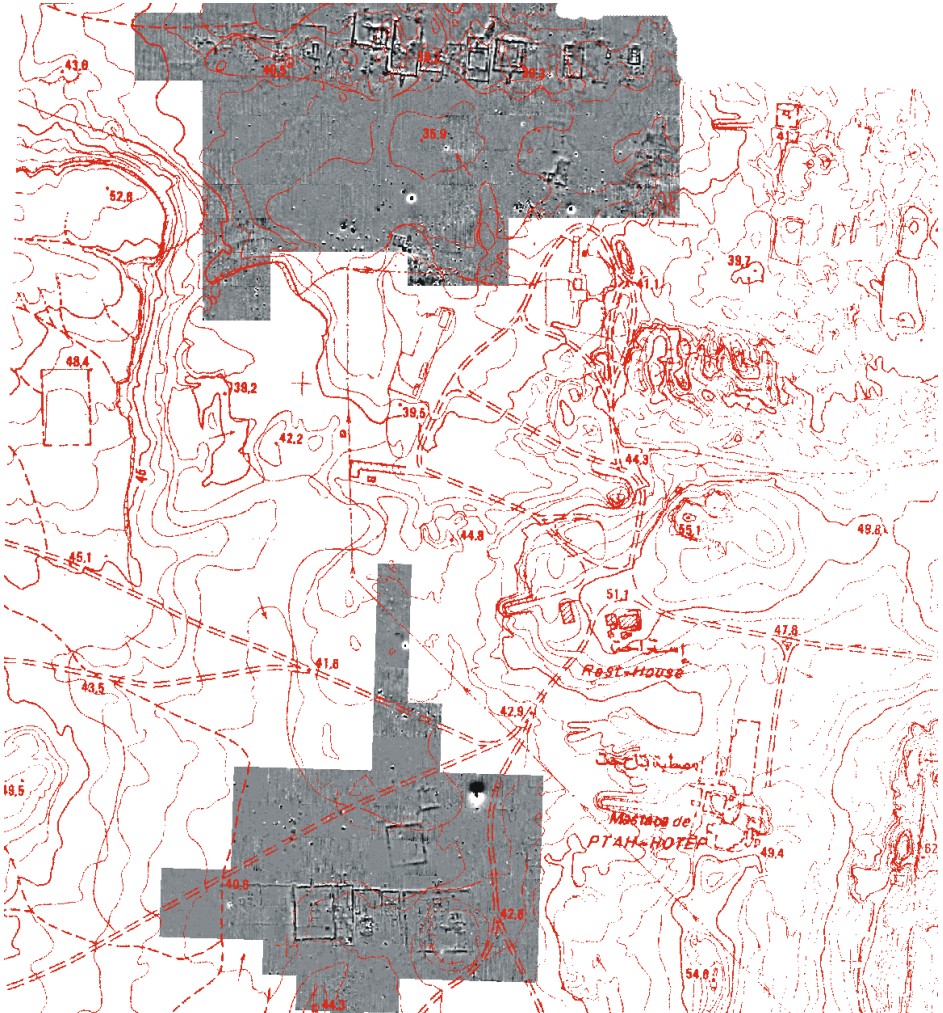

**Figure 4.** Geophysical results of the area of the Serapeum, showing two 'terraces' of temples oriented towards it, to the north and south. Based on pottery, these date to the First Millennium BCE.

## 7. Summary and Conclusions

The site of Saqqara saw intense activity at the instigation of Prince Khaemwaset, following a model set by, or which he himself helped set for, his father Ramesses II. The prince actively mobilised his relationship with the divine (great gods, dead kings, his living divinised father, and venerated non-royals) at Saqqara through monumental expressions of his royal lineage to suitably—i.e., within the confines of decorum—qualify himself for future cultic action; he planned his own divinisation, targeting both the living and eternity. This approach has often been romanticised as a sentimental 'love of antiquity' by Egyptologists who wish to see ourselves reflected in the figure of Khaemwaset. Rather, this strategy was a monumental means of leveraging his unique position, knowledge and access to resources to optimally establish his own presence in the sacred Memphite landscape.

The characterisation of Khaemwaset's works as either 'restoration' or 'recycling' betrays a generally rather limited Egyptological attitude to materiality. Presence is effected through distributing the self—in name, image and object—as widely as possible. Mindful of a range of audiences and the fate of some his predecessors, Khaemwaset was clearly keen to be remembered through his engagement with his own heritage, by re-inscribing older monuments, creating new ones from older parts and depositing large numbers of votive 'shabti' figurines throughout the Memphite necropolis.

Strangely, the canonisation of the figure of Khaemwaset in modern Egyptology has not been accompanied by a corresponding interrogation of his posthumous legacy among

the ancient Egyptians. Gomaà, among others, failed to recognise the true date of a crucial piece of evidence—the Cairo relief of the prince—and dismissed other objects associated with him that had a suggested Late Period date [1] (pp. 70–74). Khaemwaset's posthumous reputation has been largely influenced by his somewhat ribald appearance in the Demotic Setne tales. Clearly, there was an appreciation of the potency of his name, effective as guarantor to the efficaciousness of spells in late versions of the Book of the Dead, which situate his discovery of effective knowledge within the Memphite necropolis. Although, in the words of Michel Chauveau, 'no particular place of worship, no dedication of a statue or votive object indicates that Khâemouaset could have been considered the equal of a divinity' (writer's translation from the French) [2] (p. 290), the establishment of active commemoration—even if not a full cult—for him in the First Millennium BCE, most likely to have been centred at Saqqara, must now be viewed as highly plausible.

**Funding:** This research received no external funding.

**Institutional Review Board Statement:** Not applicable.

**Informed Consent Statement:** Not applicable.

**Data Availability Statement:** Not applicable.

**Acknowledgments:** This paper draws on fieldwork conducted by the writer as part of the Saqqara Geophysical Survey Project, for which opportunity I gratefully acknowledge the project and its funders, and in particular, Ian Mathieson and Jon Dittmer. I am grateful to Filippo Mi, Penny Wilson and Aidan Dodson for discussion and references, to Steven Snape for comments on a draft and to anonymous reviewers for constructive suggestions.

**Conflicts of Interest:** The author declares no conflict of interest.

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
