# Peer review of "The Legacy of Prince Khaemwaset at Saqqara"

_heritage, doi:10.3390/heritage5030115_

Round 1

Reviewer 1 Report

This is a very well written piece of work which looks at an interesting topic in an original and engaging manner.

I have only two suggestions, first that dates for Mariette, Mathieson and other Egyptologists be provided (as they have been for Norden) and second that - if possible - a map be included showing the locations of the monuments mentioned at Saqqara since not all of these will be known to a general audience.

Author Response

Many thanks for these comments. I will add dates for the Egyptologists mentioned, and attempt to source a map! 

Reviewer 2 Report

A very nice piece of work. It is high time that Khaemweset is formally contextualised for his later celebrity in print, moving these discussions from the classroom and the coffee tables!

Author Response

Thank you for these comments - I heartily agree!

Reviewer 3 Report

No further comments for author.

The paper is a concise and well-argued contribution to an important topic concerning the sacred landscape of ancient Egypt

Author Response

Thanks for these comments!